# Nonreciprocity and magnetic-free isolation based on optomechanical interactions

Freek Ruesink[1], Mohammad-Ali Miri[2], Andrea Alù[2] & Ewold Verhagen[1]

Nonreciprocal components, such as isolators and circulators, provide highly desirable functionalities for optical circuitry. This motivates the active investigation of mechanisms that break reciprocity, and pose alternatives to magneto-optic effects in on-chip systems. In this work, we use optomechanical interactions to strongly break reciprocity in a compact system. We derive minimal requirements to create nonreciprocity in a wide class of systems that couple two optical modes to a mechanical mode, highlighting the importance of optically biasing the modes at a controlled phase difference. We realize these principles in a silica microtoroid optomechanical resonator and use quantitative heterodyne spectroscopy to demonstrate up to 10 dB optical isolation at telecom wavelengths. We show that nonreciprocal transmission is preserved for nondegenerate modes, and demonstrate nonreciprocal parametric amplification. These results open a route to exploiting various nonreciprocal effects in optomechanical systems in different electromagnetic and mechanical frequency regimes, including optomechanical metamaterials with topologically non-trivial properties.

[1] Center for Nanophotonics, FOM Institute AMOLF, Science Park 104, 1098 XG Amsterdam, The Netherlands. [2] Department of Electrical and Computer Engineering, The University of Texas at Austin, Austin, Texas 78712, USA. Correspondence and requests for materials should be addressed to E.V. (email: verhagen@amolf.nl).

Lorentz reciprocity stipulates that electromagnetic wave transmission is invariant under a switch of source and observer[1], and its implications widely permeate physics. To violate reciprocity and obtain asymmetric transmission, suitable forms of time-reversal symmetry breaking are required[2]. In optical and microwave systems this is usually achieved using magneto-optic material responses. However, a vibrant search for alternative methods to break reciprocity, mimicking a magnetic bias, has taken shape in recent years[3–13]. This is fuelled by the typically weak magneto-optic coefficients in natural materials and/or their associated losses, and the technological promise of integrated on-chip nonreciprocal devices[14], including isolators and circulators. A promising approach relies on spatiotemporal modulation of the refractive index to break time-reversal symmetry. Such modulation allows imparting a nonreciprocal phase on the transfer of a signal between two optical modes[6,15] or establishing a form of angular momentum biasing to create nonreciprocity[4,16,17].

Pronounced optical time-modulation can be realized in cavity optomechanics[18,19], where the displacement $x$ of a mechanical resonator alters the resonance frequency $\omega_c$ of an optical cavity[20]. Simultaneously, light can control the mechanical motion through radiation pressure, surpassing the need for external modulation. In recent years, these interaction dynamics have been exploited for mechanical cooling[21–23], optical amplification[24], wavelength conversion[25–27] and optomechanically induced transparency[28] (OMIT). Hafezi and Rabl[29] theoretically predicted that optomechanical interactions in ring resonators can enable nonreciprocal responses, and associated asymmetric cavity spectra were recently observed[10,11,30]. In other recent work, it was recognized that the mechanically-mediated signal transfer between two optical modes can be made nonreciprocal with suitable optical driving[31,32], a mechanism that enables phonon circulators and networks with topological phases for sound and light[31,33,34].

Here we show that all of the above systems can be understood from a single description involving two optical modes coupled to a joint mechanical mode. This allows the definition of minimal conditions to achieve ideal optomechanical nonreciprocity, that is, a nonreciprocal phase shift of $\pi$ or unity isolation with vanishing insertion loss in any optomechanical system. As we show in the following, optimal nonreciprocity requires (1) driving the optical modes with a $\pi/2$ phase difference and (2) an asymmetry between the optical modes with respect to the output ports. Experimental results obtained on a ring resonator system that meets these minimal conditions are presented, showing the on-chip implementation of an optical isolator and demonstration of a nonreciprocal optomechanical amplifier.

## Results

**Nonreciprocal mode transfer and optomechanical isolation.** Consider a basic system (Fig. 1a) of two optical modes with frequencies $(\omega_1, \omega_2)$, both coupled to a mechanical mode with frequency $\Omega_m$[35]. The Hamiltonian of this system is[20]

$$H = \hbar \sum_{j=1,2} \omega_j(x) a_j^\dagger a_j + \hbar \Omega_m b^\dagger b, \qquad (1)$$

where $a$ and $b$ denote the photon and phonon annihilation operators, respectively, and $\omega_j(x) = \bar{\omega}_j - G_j x = \bar{\omega}_j - G_j x_{zpf}(b + b^\dagger)$, with $x_{zpf}$ the mechanical zero-point motion and $G_j$ the optical frequency shift per unit displacement. If both optical modes are driven by a strong coherent laser to an intracavity field $\alpha_j \exp(-i\omega_{control} t)$, the linearized Hamiltonian in a frame rotating at $\omega_{control}$ reads

$$
\begin{aligned}
H_L = &-\hbar \sum_{j=1,2} \bar{\Delta}_j \delta a_j^\dagger \delta a_j + \hbar \Omega_m b^\dagger b \\
&- \hbar \sum_{j=1,2} \left( g_j^* \delta a_j b^\dagger + g_j \delta a_j^\dagger b + g_j^* \delta a_j b + g_j \delta a_j^\dagger b^\dagger \right),
\end{aligned} \qquad (2)
$$

where $\bar{\Delta}_j = \omega_{control} - \bar{\omega}_j + G_j \bar{x}$ is the control detuning from the cavity frequency (shifted by the mean displacement $\bar{x}$), and $\delta a_j$ and $\delta a_j^\dagger$ describe the small amplitude changes of the optical field. The interaction terms on the right describe coupling between the optical and mechanical modes at rates $g_j = G_j x_{zpf} \alpha_j$, controlled through the fields $\alpha_j$.

The crucial role of the relative phases of $g_j$ is immediately revealed when considering energy-conserving pairs that mediate inter-mode transfer. For example, photon annihilation in mode 1 upon phonon creation ($g_1^* \delta a_1 b^\dagger$), and the subsequent annihilation of the phonon with photon creation in mode 2 ($g_2 \delta a_2^\dagger b$) leads to a phase pickup $\Delta\phi = \arg(g_2) - \arg(g_1)$, whereas the reverse process provides an opposite phase $-\Delta\phi$ (Fig. 1b)[31,32]. Strongest nonreciprocity is thus achieved when the two optical modes are driven with a phase difference $\Delta\phi = \pi/2$.

Interestingly, this requirement is readily met in ring resonators, such as the silica microtoroid[36] studied here. This well-known optomechanical system supports a mechanical breathing mode coupled to an even and an odd optical mode (Fig. 1c)[37]. A control beam incident through an evanescently coupled waveguide

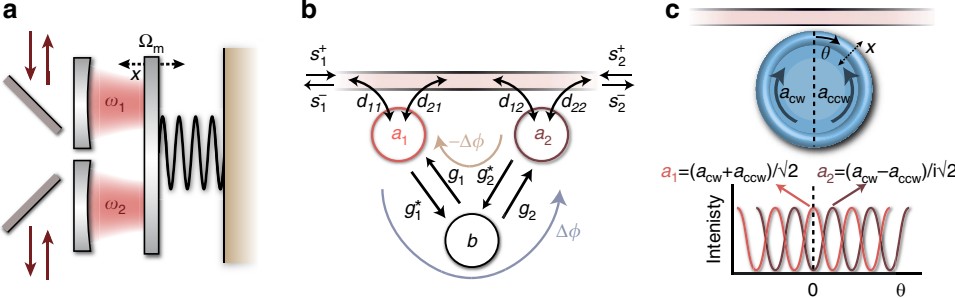

**Figure 1 | Nonreciprocity in a multimode optomechanical system.** (**a**) Optomechanical system: two optical resonators at frequencies $\omega_1$ and $\omega_2$, both coupled to a mechanical mode at frequency $\Omega_m$. (**b**) General description: the optical modes ($a_1,a_2$) are coupled to a mechanical mode $b$ with coupling rates $g_1$ and $g_2$. The path $a_1 \to b \to a_2$ picks up a phase $\Delta\phi = \arg(g_2) - \arg(g_1)$ that is opposite to that of the reversed path $a_2 \to b \to a_1$. Two input/output ports ($s_1$ and $s_2$) are coupled to the optical modes with rates $d_{ij}$. Interfering both paths with direct scattering through the waveguide can build an optical isolator. (**c**) A ring-resonator supports even and odd optical modes ($a_1$, $a_2$), superpositions of clockwise ($a_{cw}$) and counter-clokwise ($a_{ccw}$) propagating modes. As the two modes are $\pi/2$ out of phase with respect to a wave propagating in the waveguide, a control incident from a single input port fulfils the optimal driving conditions to break reciprocity. The graph sketches the spatial intensity profile of the two modes along the rim of the ring-resonator as a function of the angle $\theta$ with respect to the dashed line.

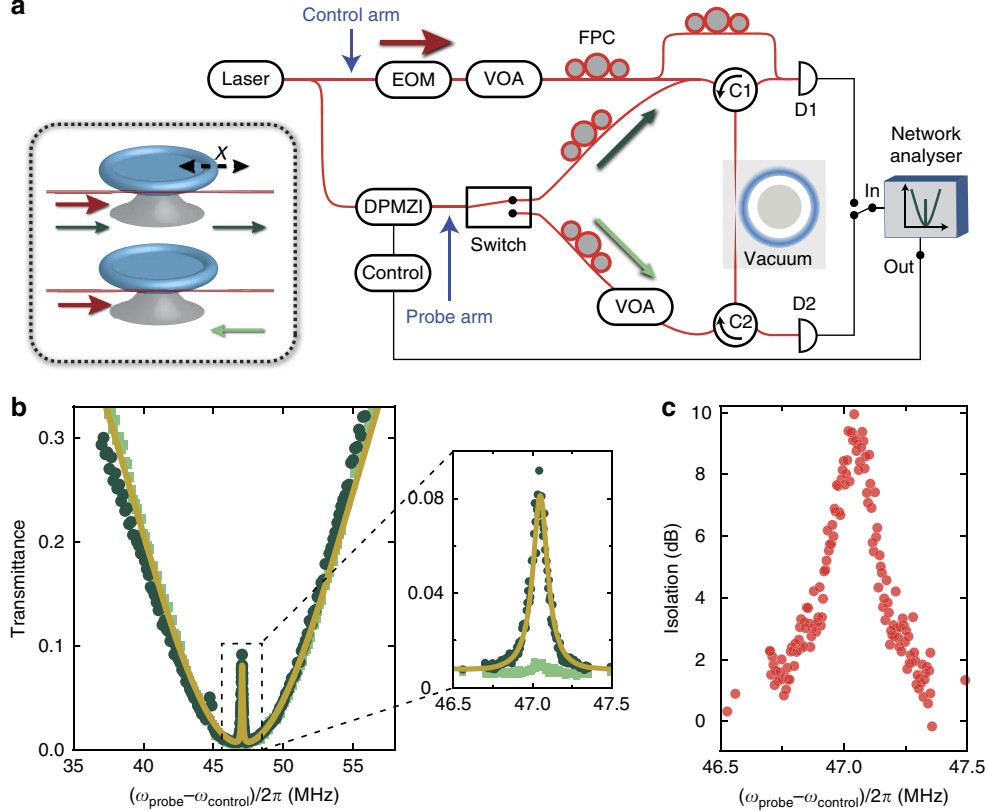

**Figure 2 | Experimental set-up and isolation.** (**a**) A fibre-coupled laser signal is split into a control and probe arm, where the probe frequency is controlled through the output of a vector network analyser (VNA) and a double-parallel Mach–Zehnder interferometer (DPMZI). An optical switch launches the probe into a tapered fibre either co-propagating or counter-propagating to the control beam. In both situations the transmittance at the probe frequency is extracted by analysing the beat with a calibrated control field using the VNA. Electro-optic modulator (EOM), variable optical attenuator (VOA), fibre polarization controller (FPC), detector (D), circulator (C). The inset sketches the pump (red arrow) and probe (green arrows) configurations when control and probe beam co-propagate (top) and counter-propagate (bottom). (**b**) Transmittance of the optical probe beam as a function of probe-control detuning with the control frequency (power ∼17 μW) tuned to the red mechanical sideband. When the probe beam co-propagates (dark green circles) with the control beam, an OMIT transmission window is observed, which is absent when the control and probe counter-propagate (light green squares), resulting in nonreciprocal optical transmission. The solid yellow line is a fit of $|S_{21}|^2$ (see Methods) using independently determined values $(\Omega_m, \Gamma_m)/2\pi \approx (47.04\,\text{MHz},$ 87.7 kHz), yielding $(\kappa_{1,2}, |g_1|)/2\pi = (28\,\text{MHz}, 292\,\text{kHz})$ and $\eta_{1,2} \approx 0.45$. (**c**) The resulting isolation, quantified as the ratio of measured probe transmittance in both directions.

excites an equal superposition of even and odd modes with $\pi/2$ phase difference[38], such that the requirement on the control phase to maximally break reciprocity is automatically fulfilled. Note that our choice of the even/odd basis (in contrast to the clockwise/counterclockwise basis considered in other work[10,29,30]) immediately reveals the role of a nonreciprocal phase in intermode coupling, unifying the description of ring resonators and other systems.

The nature of the nonreciprocal response is determined by the direct coupling between the two channels: if it is forbidden (Fig. 1a), the system primarily functions as a nonreciprocal phase shifter. If a direct pathway exists (Fig. 1c), its interference with the resonant path that collects a nonreciprocal $\pi$ phase shift enables ideal isolation under appropriate conditions. In our experiment, we demonstrate optical isolation by studying the two-way transmittance of a probe signal at frequency $\omega_{probe}$ through a tapered fibre that is coupled to a microcavity ($\omega_{1,2}/2\pi = 194.5$ THz) with linewidth $\kappa/2\pi = 28$ MHz. With the control laser incident from one direction, the transmittance is quantified using a heterodyne spectroscopic technique, where a probe beam propagating in the forward or backward direction is recombined with the control, and their beat analysed (see Fig. 2a and Methods). The fact that the measurement technique used here allows to quantify the

resulting transmittance provides a means to extract the obtained optical isolation, in contrast to the qualitative measurements reported in[10,30].

The resulting probe transmittance (Fig. 2b) for $\bar{\Delta}_{1,2} = -\Omega_m$ and near-critical coupling conditions shows a bidirectional transmission dip as the probe frequency is scanned across the cavity resonance. Importantly, the OMIT window[28], which results from destructive intracavity interference of anti-Stokes scattering of the control beam from the probe-induced mechanical vibrations with the probe beam itself, is solely present for co-propagating control and probe (dark green circles). For reversed probe direction OMIT is absent (light green squares). The device thus acts as an optical isolator, reaching up to 10 dB of isolation (Fig. 2c).

**The nonreciprocal scattering matrix.** To predict the magnitude of such nonreciprocal transmission, we use temporal coupled mode theory[39] to formulate the scattering matrix $S$ of a general system described by equation (2), relating input $(\delta s_j^+)$ and output $(\delta s_j^-)$ waves at frequency $\omega_{probe}$ in the ports $j = 1, 2$ via $(\delta s_1^-, \delta s_2^-)^{\text{T}} = S(\omega_{probe}) (\delta s_1^+, \delta s_2^+)^{\text{T}}$. The dynamics of a two-mode system described by a linear time-evolution operator

$\mathcal{M}$ reads

$$\frac{\mathrm{d}}{\mathrm{d}t}\begin{pmatrix}\delta a_1\\ \delta a_2\end{pmatrix} = \mathrm{i}\mathcal{M}\begin{pmatrix}\delta a_1\\ \delta a_2\end{pmatrix} + D^{\mathrm{T}}\begin{pmatrix}\delta s_1^+\\ \delta s_2^+\end{pmatrix}, \qquad (3)$$

where $D$ describes the mutual coupling to the input/output fields. The output fields are found from

$$\begin{pmatrix}\delta s_1^-\\ \delta s_2^-\end{pmatrix} = C\begin{pmatrix}\delta s_1^+\\ \delta s_2^+\end{pmatrix} + D\begin{pmatrix}\delta a_1\\ \delta a_2\end{pmatrix}, \qquad (4)$$

where $C$ describes the direct coupling between the two ports. Note that these expressions can be related to the quantum optics input/output formalism via a redefinition of the input fields (Supplementary Notes 1, 2 and 6). Here we prescribe the individual optical modes to be reciprocal, such that coupling to in- and outgoing fields is identical[39]. In our system, it necessitates the choice of the even/odd mode basis. In the frequency domain, equations (3 and 4) yield the total scattering matrix

$$S = C + \mathrm{i}D(M + \omega I)^{-1}D^{\mathrm{T}}, \qquad (5)$$

with $I$ the identity matrix, $M$ the Fourier transform of operator $\mathcal{M}$, and $\omega = \omega_{\mathrm{probe}} - \omega_{\mathrm{control}}$. In a general two-mode system, the difference between forward and backward complex transmission coefficients thus reads

$$S_{21} - S_{12} = \frac{\mathrm{i}\,\det(D)(m_{12} - m_{21})}{\det(M + \omega I)}, \qquad (6)$$

showing that reciprocity can be broken as long as $\det(D) \neq 0$ and $m_{12} \neq m_{21}$ (with $m_{ij}$ the elements of $M$). This important result identifies the minimal conditions to break reciprocity: a full-rank $D$ matrix, requiring an asymmetry in the coupling between the two optical modes and the channels $s_1$ and $s_2$, and an asymmetric evolution matrix, enforcing the coupling rate from mode 1 to mode 2 to be different from that of mode 2 to mode 1. As explained above, this can be implemented through optomechanical interactions.

The evolution matrix $M$ that describes optomechanical interactions (Fig. 1) is obtained from the equations of motion

$$\frac{\mathrm{d}}{\mathrm{d}t}\begin{pmatrix}\delta a_1\\ \delta a_2\end{pmatrix} = \mathrm{i}\begin{pmatrix}\bar{\Delta}_1 + \mathrm{i}\kappa_1/2 & 0\\ 0 & \bar{\Delta}_2 + \mathrm{i}\kappa_2/2\end{pmatrix}\begin{pmatrix}\delta a_1\\ \delta a_2\end{pmatrix}$$
$$+ \mathrm{i}\begin{pmatrix}g_1(b + b^{\dagger})\\ g_2(b + b^{\dagger})\end{pmatrix} + D^{\mathrm{T}}\begin{pmatrix}\delta s_1^+\\ \delta s_2^+\end{pmatrix}, \qquad (7)$$

$$\frac{\mathrm{d}}{\mathrm{d}t}b = (-\mathrm{i}\Omega_{\mathrm{m}} - \Gamma_{\mathrm{m}}/2)b + \mathrm{i}(g_1^* \delta a_1 + g_1 \delta a_1^{\dagger} + g_2^* \delta a_2 + g_2 \delta a_2^{\dagger})$$
$$+ \sqrt{\Gamma_{\mathrm{m}}}\,b_{\mathrm{in}}, \qquad (8)$$

derived from the linearized Hamiltonian (2) including dissipation and coupling between the mechanical resonator and its thermal bath (rightmost term in equation (8)), which under the experimental conditions studied here can be ignored in the analysis (Methods). Likewise we neglect optical quantum noise. Note that we have set coupling between the optical modes to zero, which can always be realized through diagonalization (see Supplementary Note 4). Solving these equations in the frequency domain, applying the rotating wave approximation and using the input-output relation (4), the evolution matrix $(M + \omega I)$ for $\omega \approx \pm \Omega_{\mathrm{m}}$ reads

$$M + \omega I = \begin{pmatrix}\Sigma_{\mathrm{o}_1} \mp \frac{|g_1|^2}{\Sigma_{\mathrm{m}}^{\pm}} & \mp \frac{g_1 g_2^*}{\Sigma_{\mathrm{m}}^{\pm}}\\ \mp \frac{g_1^* g_2}{\Sigma_{\mathrm{m}}^{\pm}} & \Sigma_{\mathrm{o}_2} \mp \frac{|g_2|^2}{\Sigma_{\mathrm{m}}^{\pm}}\end{pmatrix}. \qquad (9)$$

Here, $\Sigma_{\mathrm{o}_j} = \omega + \bar{\Delta}_j + \mathrm{i}\kappa_j/2$ is the inverse optical susceptibility, $\Sigma_{\mathrm{m}}^{\pm} = \omega \mp \Omega_{\mathrm{m}} + \mathrm{i}\Gamma_{\mathrm{m}}/2$ the inverse mechanical susceptibility and $\Gamma_{\mathrm{m}}$ the mechanical damping rate. Importantly, $(m_{12} - m_{21}) \propto \sin\Delta\phi$, highlighting the importance of the control phase difference to obtain nonreciprocal transmission.

We define individual cooperativities $\mathcal{C}_j$ by $\mathcal{C}_j \equiv 4|g_j|^2/(\kappa_j \Gamma_{\mathrm{m}})$ and the total cooperativity $\mathcal{C} \equiv \mathcal{C}_1 + \mathcal{C}_2$. By combining (6) and (9), the asymmetric transmission through a two-mode system

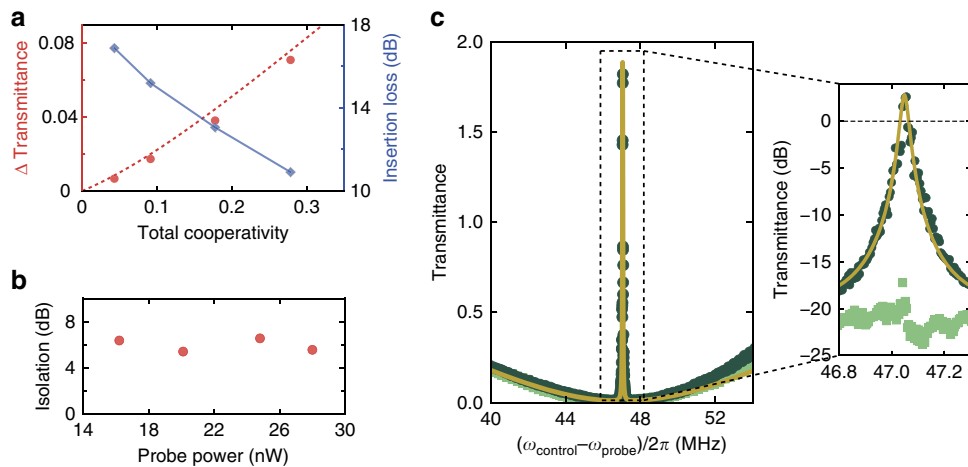

**Figure 3 | Power-dependence and nonreciprocal amplification. (a)** Difference between forward/backward transmissivities (measured, red circles and theory ($|S_{21}|^2 - |S_{12}|^2$), dashed red line) with respect to cooperativity, directly proportional to the control laser power. Together with an increase in contrast, the insertion losses (blue diamonds) decrease with increasing cooperativity **(b)** The isolation as a function of probe power sent through the fibre. The physical mechanism behind optical isolation is linear, and thus does not depend on probe power. **(c)** When the control beam is tuned to the blue side band of the cavity, it can parametrically amplify the probe beam that co-propagates with it through the fibre (dark green circles). In contrast, the counter-propagating probe beam (light green squares) experiences normal cavity extinction, thus yielding a nonreciprocal amplifier. With amplification of $\sim 3\,\mathrm{dB}$, the nonreciprocal difference in transmission is $\sim 23\,\mathrm{dB}$. The solid yellow line is a fit of $|S_{21}|^2$ (see Methods) yielding $(\kappa_{1,2}, |g_1|)/2\pi \approx (28\,\mathrm{MHz}, 454\,\mathrm{kHz})$ and $\eta_{1,2} \approx 0.46$.

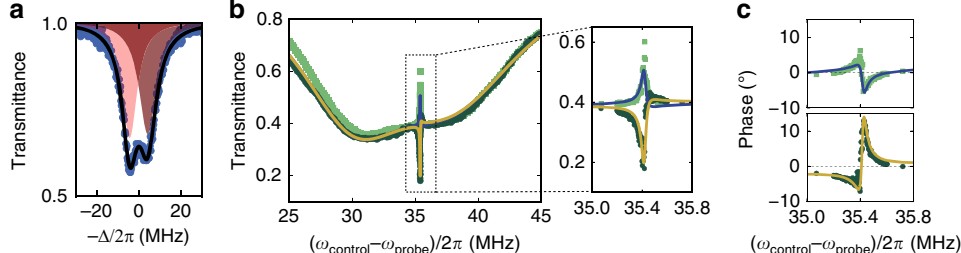

**Figure 4 | Non-degenerate optical modes.** (**a**) Transmittance of an optical split-mode (splitting ~8.6 MHz) as a function of laser-cavity detuning, obtained by sweeping the laser frequency and measuring the resulting transmittance using an oscilloscope. The horizontal axis is calibrated using the EOM placed in the control arm (see Fig. 2a). The black solid line represents a fit of a double Lorentzian lineshape to the blue data points. The red shading is the area under the two fitted Lorentzian lineshapes. (**b**) Transmittance of the optical probe beam as a function of control-probe detuning with the control frequency fixed at the blue mechanical side band. When the probe beam co-propagates ($|S_{21}|^2$, dark green circles) with the control beam an optomechanically induced absorption window appears, while the oppositely propagating probe ($|S_{12}|^2$, light green squares) experiences increased transmission. (**c**) Asymmetric phase transmission for the same measurement as **b**. Light green squares correspond to $\arg(S_{12})$, the dark green circles to $\arg(S_{21})$. The solid lines in **b** are fitted simultaneously to $|S_{12}|^2$ (blue line) and $|S_{21}|^2$ (yellow line). The resulting parameters are inserted in $S_{12}$ (blue line) and $S_{21}$ (yellow line) to yield the lineshapes in **c** (Methods).

can be written (Supplementary Notes 3 and 5) as

$$S_{21} - S_{12} = 2\sin\Delta\phi\sqrt{\eta_1\eta_2} \times$$
$$\frac{\mp 2\sqrt{\mathcal{C}_1\mathcal{C}_2}}{(\delta_\pm + i)(\delta_1 + i)(\delta_2 + i) \mp (\mathcal{C}_2(\delta_1 + i) + \mathcal{C}_1(\delta_2 + i))}, \quad (10)$$

where we defined the relative detuning of the probe frequency $\delta_\pm \equiv (\omega \mp \Omega_\mathrm{m})/(\Gamma_\mathrm{m}/2)$ and $\delta_j \equiv (\omega + \bar{\Delta}_j)/(\kappa_j/2)$ from the mechanical and optical resonance, respectively, and $\eta_j$ is the fraction of energy mode $j$ radiates in both output channels. Inspection of equation (10) shows that the magnitude of asymmetric transmission at critical coupling ($\eta_{1,2} = 1/2$) is maximally 1, when the cooperativities are large and equal. These conditions, implemented in our experiment, enable the observed strong optical isolation.

**Dependence on power and detuning and mode degeneracy.** For degenerate optical modes and the control field tuned to either mechanical sideband, the maximum contrast between forward and backward transmittance is $\Delta T = |S_{21}|^2 - |S_{12}|^2 \approx (\mathcal{C}^{-1} \pm 1)^{-2}$ at $(\omega = \pm\Omega_\mathrm{m}, \bar{\Delta}_{1,2} = \mp\Omega_\mathrm{m})$, where $\mathcal{C} = 2\mathcal{C}_1 = 2\mathcal{C}_2$. The pronounced increase of $\Delta T$ with increasing $\mathcal{C}$, and concomitant decrease of insertion loss, are confirmed by varying the optical drive power (Fig. 3a). The mechanism has strong potential for near-ideal isolation at negligible insertion losses, for example in optimized silica microtoroids, where $\mathcal{C} \approx 500$ was demonstrated[23]. Moreover, cooperativity enhances the bandwidth, which is ultimately limited by the optical linewidth[29]. An important aspect of this mechanism is that the isolation is independent of probe power (Fig. 3b), differing fundamentally from mechanisms exploiting static nonlinearity[40,41] to create asymmetric transmission. Note that noise photons originating from the mechanical thermal bath contribute only 0.4% to the measured probe signal (Methods).

For blue-detuned control $(\bar{\Delta}_{1,2} = +\Omega_\mathrm{m})$, the probe beam experiences parametric amplification if control and probe are co-propagating, while it is fully dissipated when counter-propagating with the pump, thus yielding a nonreciprocal optical amplifier (Fig. 3c). This feature could pose interesting signal processing functionality, including nonreciprocal narrowband RF filtering and insertion loss compensation.

Importantly, equation (10) shows that strong nonreciprocity can also be obtained without optical degeneracy. If the two modes have different frequency and/or linewidth, an optimal control frequency can be chosen to satisfy $\delta_1 = -\delta_2 = \beta$. Then

asymmetric transmittance is maximally

$$\Delta T = \frac{\mathcal{C}(\mathcal{C} \pm 2\beta^2)}{(1 \pm \mathcal{C} + \beta^2)^2}, \quad (11)$$

showing that larger cooperativity can compensate the effects of mode splitting for $\beta > 1$. Figure 4 shows nonreciprocal amplitude and phase transmission with a split optical mode. A probe beam tuned between the even and odd mode frequencies excites both modes with unequal phases. These opposing phases are added to the $a_1 \to a_2$ and $a_2 \to a_1$ optomechanical mode conversion processes, respectively, changing the interference condition with the nonresonant transmission. As a result, both co- and counter-propagating probe fields now interact with the mechanical mode. For a blue-detuned control beam, this yields induced absorption for the co-propagating probe and induced transparency for the counter-propagating probe (Fig. 4b). Note that the induced absorption for the co-propagating beam is related to the relatively low coupling rate ($\eta_{1,2} < 0.5$). It can be turned into gain, as presented in Fig. 3c, for $\eta_{1,2} > 0.5$ and/or for increased optical control power. Crucially, since for our system the relation $\beta \ll \Omega_\mathrm{m}/\kappa_{1,2}$ holds (Methods), the deviation of $\Delta\phi$ from optimal is only 0.2%. As such, a control beam incident from one side still ensures $\Delta\phi \approx \pi/2$ and $\mathcal{C}_1 \approx \mathcal{C}_2$, thus fulfilling the requirements for optimal nonreciprocity and maximizing the contrast between forward and backward transmission. In a more general case, optimal conditions may be implemented, for example by supplying control fields with suitable phase and amplitude through both input waveguides. Importantly, the fact that nonreciprocity can be obtained without optical degeneracy increases the range of systems that may be employed.

**Discussion**
We stress that the demonstrated principles are not limited to the experimental implementation using ring resonators shown here, but can be realized in a wide range of optomechanical platforms[20], such as LC circuits[27] and photonic crystal resonators[22,26] (Fig. 5a,b). In fact, the high (GHz) frequency of such devices has the prospect of enhancing the bandwidth with respect to the relatively narrow range demonstrated here, towards a range commensurate with typical signal modulation rates. While the resonant nature of the demonstrated mechanism is of course a limit to the general application capability, we foresee several applications that could benefit from magnetic-free isolation over a finite bandwidth. These include in particular

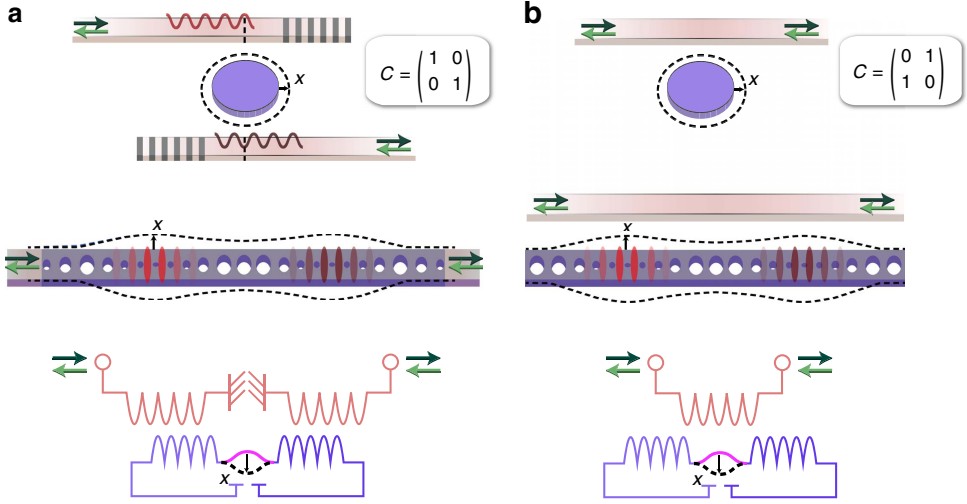

**Figure 5 | General optomechanical nonreciprocity in different systems.** (**a**) If the optical and mechanical resonators are placed in the direct propagation path, the displayed systems have a diagonal direct scattering matrix $C$, which builds a nonreciprocal phase shifter. (**b**) However, if the optomechanical system provides an extra (nonreciprocal) propagation path in addition to the direct scattering path, the $C$ matrix has off-diagonal elements which can interfere with the nonreciprocal path to yield isolation or amplification. Here, Dark and light green arrows represent wave transmission from opposite system ports.

the protection of on-chip monochromatic laser sources and, with ground-state cooling[21,22] or in the strong coupling regime[23], low-loss routing of signals carrying quantum information at negligible added noise, either at optical or microwave frequencies[13,29].

The specific nonreciprocal functionality is governed by the way these systems are coupled to input/output channels. This, in turn, is directly related to the nonresonant scattering matrix $C$, as reciprocity of optical modes dictates $CD^* = -D$ (ref. 39). For the scenarios in Fig. 5a, described by a diagonal $C$ matrix, each waveguide couples to a single optical mode, and the system operates as a nonreciprocal phase shifter (gyrator) in the absence of other optical loss. Importantly, an on-chip gyrator that is placed in one arm of an integrated Mach-Zehnder interferometer could be used to build an on-chip circulator[42]. In contrast, isolation is most naturally achieved if $C$ is the exchange matrix, meaning a direct path between the two ports is present (Fig. 5b). We note that nonreciprocity occurs also outside the resolved side-band regime, although the behaviour there is more complex due to mixing of sidebands at $\pm\omega$.

In conclusion, we demonstrated and quantified nonreciprocal transmission through a compact optomechanical isolator and parametric amplifier, and developed a general theory explaining the mechanism and unifying the description of various implementations of optomechanical nonreciprocity in multimode systems. Our findings identify two general requirements for any optomechanical system to optimally break reciprocity: asymmetric coupling of the optical modes to input/output channels, and a drive phase-difference of $\pi/2$. Since the requirements for optimal nonreciprocity derived here do not rely on the handedness of optical[29,30] or mechanical[10,11] modes, our theoretical formalism can be used to realize optomechanical nonreciprocity in systems that do not exhibit circular symmetry (Fig. 5). Extending the demonstrated principles to more modes or channels would enable a variety of nonreciprocal functionality for both light and sound, including on-chip circulation, gyration[31] and enhanced isolation bandwidth. Finally, these nonreciprocal systems can form the unit cell of optomechanical metamaterials with topologically non-trivial properties, where the nonreciprocal phase takes the role of an effective gauge field to establish new phases for sound and light[33,34].

*Note added in proof*: After submission, we became aware of related work by Fang *et al.*[43] that reports nonreciprocal transmission in an optomechanical crystal circuit that relies on the same principle with blue-detuned control.

## Methods

**Coupling matrix and drive condition in ring resonator.** Time-reversal symmetry and energy conservation dictate that $CD^* = -D$ and $D^\dagger D = \mathrm{diag}(\eta_1\kappa_1, \eta_2\kappa_2)$[39]. Applying these to the even and odd optical modes ($\delta a_1, \delta a_2$) of an evanescently coupled ring resonator, and choosing $c_{21} = c_{12} = 1$, constrains the coupling matrix $D$ to

$$D = \frac{1}{\sqrt{2}}\begin{pmatrix} \mathrm{i}\sqrt{\eta_1\kappa_1} & -\sqrt{\eta_2\kappa_2} \\ \mathrm{i}\sqrt{\eta_1\kappa_1} & \sqrt{\eta_2\kappa_2} \end{pmatrix}. \quad (12)$$

Together with equations (5 and 9), this $D$ matrix yields the complete expressions for the scattering matrix elements

$$S_{ij} = c_{ij} + \mathrm{i}\frac{2A_{ij} \pm 2\sqrt{\mathcal{C}_i\mathcal{C}_j}\left(d_{ij}d_{ji}\mathrm{e}^{\mathrm{i}(\phi_j - \phi_i)} + d_{ii}d_{jj}\mathrm{e}^{\mathrm{i}(\phi_i - \phi_j)}\right)}{\sqrt{\kappa_i\kappa_j}\left[(\delta_\pm + \mathrm{i})(\delta_i + \mathrm{i})(\delta_j + \mathrm{i}) \mp \left(\mathcal{C}_i(\delta_j + \mathrm{i}) + \mathcal{C}_j(\delta_i + \mathrm{i})\right)\right]}, \quad (13)$$

where $A_{ij}$ is given by

$$A_{ij} = d_{ii}d_{ji}\sqrt{\frac{\kappa_j}{\kappa_i}}\left[(\delta_\pm + \mathrm{i})(\delta_j + \mathrm{i}) \mp \mathcal{C}_j\right] + d_{ij}d_{jj}\sqrt{\frac{\kappa_i}{\kappa_j}}\left[(\delta_\pm + \mathrm{i})(\delta_i + \mathrm{i}) \mp \mathcal{C}_i\right], \quad (14)$$

used to fit the data in Figs 2–4.

For a single drive field with amplitude $\bar{s}_{\mathrm{control}}$ incident through port 1 and using $G_1 = G_2 = G$, the coupling rates $g_1$ and $g_2$ are given by

$$
\begin{aligned}
\begin{pmatrix} g_1 \\ g_2 \end{pmatrix} &= Gx_{\mathrm{zpf}}\begin{pmatrix} \alpha_1 \\ \alpha_2 \end{pmatrix} \\
&= Gx_{\mathrm{zpf}}\begin{pmatrix} \Sigma_{\mathrm{o}_1}^{-1}(\omega=0) & 0 \\ 0 & \Sigma_{\mathrm{o}_2}^{-1}(\omega=0) \end{pmatrix}D^{\mathrm{T}}\begin{pmatrix} \bar{s}_{\mathrm{control}} \\ 0 \end{pmatrix} \\
&= -\frac{Gx_{\mathrm{zpf}}\,\bar{s}_{\mathrm{control}}}{\sqrt{2}}\begin{pmatrix} \frac{\sqrt{\eta_1\kappa_1}}{\bar{\Delta}_1 + \mathrm{i}\kappa_1/2} \\ \mathrm{i}\frac{\sqrt{\eta_2\kappa_2}}{\bar{\Delta}_2 + \mathrm{i}\kappa_2/2} \end{pmatrix}.
\end{aligned} \quad (15)
$$

Thus for large detuning $|\bar{\Delta}_{1,2}| \gg \kappa_{1,2}$, the optimal phase difference $\Delta\phi = \pi/2$ is automatically satisfied by pumping through a single channel.

**Experimental set-up.** The silica microtoroid (diameter 41 μm) is fabricated using techniques as previously reported (see for example ref. 37). A tuneable fibre-coupled external cavity diode laser (New Focus, TLB-6728) is locked (using the electro-optic modulator) to a mechanical sideband of a whispering gallery mode at 1,542 nm using a modified Pound-Drever-Hall scheme that can be used independent of the probe beam direction. The probe light is generated using a commercial double-parallel Mach–Zehnder interferometer (Thorlabs, LN86S-FC)

operated in single-side-band carrier-suppressed mode, driven by the output of a vector network analyser (VNA) at frequency $\Omega$. The resulting probe light has frequency $\omega_{probe} = \omega_{control} \pm \Omega$. The sign of the frequency shift, as well as the suppression of the carrier (by 50 dB with respect to the generated probe) is controlled by bias voltages applied to the double-parallel Mach–Zehnder interferometer. Pump and probe amplitude and polarization are controlled with variable optical attenuators and fibre polarization controllers (Fig. 2a). The probe beam propagating in forward or backward direction is recombined with the control beam and their beat on fast (125 MHz) low-noise photo receivers (D1/D2) is analysed with a VNA. It should be noted that fluctuations of the optical length difference of probe and control paths generate phase fluctuations of the beat analysed by the VNA. To minimize these phase fluctuations on the time scale of the inverse bandwidth $(5 \, \text{kHz})^{-1}$ of the VNA, the lengths of the paths Laser/C1/D2 and Laser/Switch/C1/D2 are matched, as well as those of the paths Laser/D1 and Laser/Switch/C2/D1 (see Fig. 2a).

**Measurement procedure and fitting.** Before each measurement the probe power in both propagation directions is balanced using a variable optical attenuator in one of the probe arms. The polarization of both probe directions is controlled via fibre polarization controllers, which are tuned separately to optimize the fibre-to-cavity-mode coupling. To calibrate the transmittance at the probe frequency, a reference measurement is performed with control and probe tuned away from the cavity resonance. Both the reference and measurement are averages of 75 traces of a frequency-swept probe. For each measurement, $|S_{ij}|^2$ is fitted over a wide $\omega$ range used to determine $\bar{\Delta}_j$ and $\kappa_j$. Fixing these values, the same equation is fitted to a smaller frequency range surrounding the OMIT peak to yield values for $\eta_j$ and $|g_j|$. In all fits, $\Omega_m/2\pi$ and $\Gamma_m/2\pi$ are kept fixed at the independently determined values from thermal noise spectra obtained with a spectrum analyser. For the split-mode experiment, the fit result yields $\Omega_m/2\pi \approx 35.4 \, \text{MHz}$, $|\omega + \bar{\Delta}_{1,2}|/2\pi \approx 4 \, \text{MHz}$ and $\kappa_{1,2}/2\pi \approx 12 \, \text{MHz}$. Using these values in (15) directly gives a deviation from the optimal drive phase $\Delta\phi$ of only $\sim 0.2\%$. The solid curves in Fig. 4c are directly obtained from the fit results from Fig. 4b, with no other fit parameters than a vertically offset. The theory curve in Fig. 3a is obtained using the average value $\eta_j = 0.453$ as determined from the four measurements at different control powers.

**Noise due to the thermal bath.** In the resolved-sideband regime, for degenerate modes with equal driving and linewidth, the amount of detected photons per second ($N_{noise}$) that is generated through a coupling between the mechanical resonator and the heat bath reads

$$N_{noise} \approx \Gamma_m N_{th} \times \frac{\Gamma_{cool}}{\Gamma_{cool} + \Gamma_m} \eta \times \frac{\Delta B}{\Gamma_{cool} + \Gamma_m}, \qquad (16)$$

where $N_{th} \approx \frac{k_B T}{\hbar \Omega_m}$, $\Gamma_{cool} = \frac{4g^2}{\kappa}$ and the measurement bandwidth (in our experiment the VNA bandwidth) is given by $\Delta B$. From left to right, the terms in this equation can be associated with the number of noise photons generated in the resonator, the fraction of noise photons leaving through the optical channel, and the fraction of noise photons in the signal bandwidth, respectively. Note that the expression can be rewritten to yield $N_{noise} = (\mathcal{C} N_{th} \eta \Delta B)/(\mathcal{C}+1)^2$, from which $N_{noise} \approx 4 \times 10^8$ is obtained for our system. As a probe power of 15 nW at 1,542 nm corresponds to $\approx 1 \times 10^{11}$ photons/s, the thermal noise in our system contributes only marginally (0.4%) to the measured probe signal.

**Data availability.** The data that support the findings of this study are available from the corresponding author upon reasonable request.

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

## Acknowledgements

This work is part of the research programme of the Foundation for Fundamental Research on Matter (FOM), which is part of the Netherlands Organization for Scientific Research (NWO). It has been supported by the Office of Naval Research, and the Simons Foundation. We acknowledge F. Alpeggiani for useful comments.

## Author contributions

F.R. fabricated the samples and carried out the experiments. F.R. and E.V. developed the experimental set-up. M.-A.M. developed the theoretical model, with contributions from E.V., A.A. and F.R. F.R. and M.A.M. analysed the data. E.V. and A.A. supervised the project. All authors contributed to the writing of the manuscript.

## Additional information

**Competing financial interests:** The authors declare no competing financial interests.

**Publisher's note**: 

