## [Peer Review File · Nature Communications]

Reviewers' comments:

Reviewer #1 (Remarks to the Author):

This manuscript reports optomechanically induced nonreciprocity in a silica microtoroid resonator, which was inspired by a theory proposed by Hafezi and Ral (Ref 27). New technologies for Magnetic-free nonreciprocity are of great interest in the field of photonics. The results presented in this manuscript demonstrated clearly that nonreciprocity can be achieved in a waveguide coupled optomechanical resonator due to asymmetric coupling of a mechanical mode with optical modes propagating in different directions. The authors presented their theoretical models and demonstrated their prediction by using probe-control experiments. Their results showed that probe signals experienced optomechanically induced transparency (amplification) when control frequency was tuned to the red (blue) mechanical sideband. The manuscript is well written and the results are interesting. However, Z. Shen and his colleagues have reported similar work ("Experimental realization of optomechanically induced nonreciprocity", Z. Shen, Y.L. Zhang, Y.Che, C.L. Zou, Y.F.Xiao, X.B.Zou, F.W.Sun, G.C. Guo, C.H. Dong, Nature Photonics, 2016). To warrant the publication of their work in Nature Communications, the authors should make it clear the novelty of their results and explain the different of their work from Shen's.

In addition, here are a few other comments on this manuscript:

1. The authors suggested that their system can function as a nonreciprocal phase shifter. It will be helpful if they can present a figure showing the experimental results of nonreciprocal phase response.
2. In the bottom of page 5, the authors have some discussion about cooperativity. The difference between forward/backward transmissivities with respect to cooperativity was presented in Figure 3a. However, I couldn't find the particular definition of cooperativity in the manuscript, which makes it difficult to understand the relationship between the cooperativity and the forward/backward transmission.
3. The results presented in Figure 3 and Figure 4 are confusing. Figure 4b shows that optomechanically induced absorption appears when the control frequency is fixed at the "blue mechanical sideband". However, a different statement was presented in Figure 3c, which demonstrates that "when the control beam is tuned to the blue-sideband of the cavity, it can parametrically amplify the probe beam that co-propagates with it through the fibre."
4. Figure 4b shows that oppositely propagating probe experiences increased transmission. However, the results in previous figures indicate that the coupling between counter-propagating probe-control beams with the mechanical mode is negligible. Then what does the change in the probe come from if it propagates in opposite direction from the control beam?

Reviewer #2 (Remarks to the Author):

The authors demonstrate optical isolation of a probe wave by creating a non-reciprocal opto-mechanical interaction with a control wave mediated by a breathing mode in a silica microtoroid. In one direction of propagation the probe wave experiences OMIT which creates a transmission band while for the reverse direction of propagation the probe is attenuated. I think the experimental results are clearly presented, however, I have several concerns that the authors need to answer or to address.

(1) The basic scheme of optomechanical isolation presented here strikes me as very similar to the Bahl work (reference 10 in the manuscript). Bahl's paper uses electrostriction to set up OMIT for a probe wave coupling to a silica microsphere resonator. As in the present manuscript the OMIT allows passage of the probe in one direction (determined by the control) while the reverse direction is attenuated because selection rules prevent OMIT from occurring. In the Bahl paper,

Brillouin phonons mediate the process in the silica microsphere via electro-striction while in the present case a mechanical breathing mode mediates the process in the micro toroid via radiation pressure. The parallel between the electro-striction-based optomechanical coupling and the radiation pressure optomechanical coupling is well known for cooling (see Bahl, ... Carmon, Nature Physics (2012)). In short, even though the work presented is sound, I am not convinced that the present demonstration is distinct enough from the Bahl nonreciprocal work to merit publication in a Nature journal. The authors should provide some comment here to answer my concern. Perhaps there is also a way to address it further in the manuscript.

(2) I have a concern about the practical application of the present idea for isolation. The introductory paragraph partially motivates this work on the basis of chip-based isolation and inherent limitations of present isolators. However, the present approach has two very serious weaknesses relative to commercial isolators based on magnetic materials. First, it requires a control wave that has a well-defined frequency relative to the probe wave. Second, and even more restrictive is that the isolation bandwidth is extremely narrow. Could the authors provide an example in the discussion section of the manuscript to illustrate how the present method of isolation could be applied in some way. As an additional question, is there a geometry in which a circulator (also mentioned in the introduction) could be realized.

(3) There are several critical, early works related to the present work that have unfortunately not been cited. There is a trend in this field to cite a recent Rev. Mod. Phys. Paper (ref 18) as a means to mass cite the field. This is acceptable provided that one does not compromise precision or omit citations that are truly relevant. In the present case, a number of citations (19-21 and 23-25 for example) have little relevance to the present work and could be omitted as they are included in ref. 18. On the other hand, several works should be cited as they either helped to launch the subject of cavity optomechanics, describe the first report of the resonator used in this paper, or describe the first implementation of a dual optical mode system in which the modes interact by way of a breathing mode in a microtoroid. Below are the citations that should be included along with an explanation.

The following two papers are the first reports of the microcavity-based optomechanical interaction used in this paper. Indeed, the "pronounced optical time domain modulation" noted in the first sentence of the manuscript's second paragraph was first reported in these papers. They are also the first microcavity demonstrations of dynamic backaction and introduce the concept of mechanical gain via optomechanical coupling. Also, the microcavity system used was a microtoroid, the same as the present work.

[1] Carmon, et. al. "Temporal Behavior of Radiation-Pressure-Induced Vibrations of an Optical Microcavity Phonon Mode" Phys. Rev. Lett. 94, 223902 (2005)

[2] Kippenberg, et. al. "Analysis of Radiation-Pressure Induced Mechanical Oscillation of an Optical Microcavity" Phys. Rev. Lett. 95, 033901 (2005)

As noted by the authors, the resonator used in this work is a microtoroid and when introduced the authors should cite the original paper on this resonator

[3] Armani, et. al. "Ultra-high-Q toroid microcavity on a chip" Nature 421, 925-928 (2003)

The concept of a dual optical mode system in which the modes interact by way of a breathing mode was first demonstrated below (in a microtoroid system) and should be cited.

[4] Grudinin et. al. "Phonon laser action in a tunable two-level system" Phys. Rev. Lett. 104, 083901 (2010)

Reviewer #3 (Remarks to the Author):

This manuscript presents an experimental implementation of a method to generate an effectively non-reciprocal optical system by exploiting the optomechanical interaction between a mechanical resonator and two optical fields. This contrasts with the more common mechanism used to generate optical non-reciprocity in that it does not require the use of any magnetic fields, making it inherently more suitable for chip-scale devices.

I am confident that the experimental results in the manuscript and their interpretation are broadly correct. Moreover, the topic is of great importance nowadays, as evidenced by another recent preprint [arXiv:1608.03620] that describes the experimental realisation of a similar mechanism. For these reasons I believe that the manuscript is, /in principle/, publishable in Nat. Commun. However, I am not convinced by the mathematical methods used by the authors, as I discuss in detail below, and would like to ask them to revisit their work before resubmitting.

(1) My main technical issue with the manuscript centres around Eqs. (3) and (4). As a quantum optician, my first thought when I read the manuscript and supplementary information was that the matrices C and D can be derived by means of cavity QED and well-known input--output relations. As it turns out, these relations are inconsistent with the above-mentioned equations. Specifically, D^{\dagger} in Eq. (3) needs to be replaced by $-D^{\dagger}$ (i.e., minus the hermitian conjugate of the matrix D). On digging deeper, it looks to me like the sign can be absorbed into a redefinition of the input fields (but not the outputs); the main problem concerns the difference between the transpose and the conjugate. Upon close inspection of Eq. (2) in the authors' Ref. [1] in the supplementary information, the answer seems to be that the authors are using equations that use the "engineers' convention" for the frequency ($da/dt = i\omega a + \dots$), whereas in the manuscript the opposite convention is used ($da/dt = -i\omega a + \dots$). A derivation of D based on cavity QED and the input--output relation also seems to disagree with Eq. (S38). I encourage the authors to provide a microscopic (by which I mean: cavity QED and input--output) derivation of their C and D matrices, which I am convinced is possible, and to ensure that their mathematics is self-consistent. The model and results seem to match quite well in any case, so I do not expect major changes to result from this, but I do expect the mathematics in the manuscript and the supplementary information to be correct.

(2) In Eq. (8), b is an operator. Conservation of the canonical commutation relations requires that noise terms are added to Eq. (8). Do these noise terms have any effect? If not, they should still be included in the text for correctness. These noise terms are also missing in Eq. (S10).

I have a few other, minor, issues that would also like to see resolved:

(3) The preprint I quoted above, arXiv:1608.03620, should be referred to.

(4) Following Eq. (11) it is stated that $\beta \ll \Omega_m/\kappa_{\{1,2\}}$, but no justification is given for this assertion.

(5) In the supplementary material, Eq. (S8) presupposes that $c_{\{21\}} = c_{\{12\}}$. This is mentioned in Ref. [1] but should be clearly stated in the manuscript.

(6) Following Eq. (S10) there is a spelling mistake: "optomchanical"

(7) Following Eq. (S37) the authors do not define what they mean by "such that equations (S1) and (S2) are consistent". As is implied above, this whole discussion would be rendered moot if the authors provided a proper microscopic derivation of matrices C and D.

Reviewer #1 (Remarks to the Author):

The manuscript is well written and the results are interesting. However, Z. Shen and his colleagues have reported similar work (“Experimental realization of optomechanically induced nonreciprocity”, Z. Shen, Y.L. Zhang, Y.Che, C.L. Zou, Y.F.Xiao, X.B.Zou, F.W.Sun, G.C. Guo, C.H. Dong, Nature Photonics, 2016). To warrant the publication of their work in Nature Communications, the authors should make it clear the novelty of their results and explain the different of their work from Shen’s.

While there are clear relations to the advanced online publication in Nature Photonics (August 22, 2016) by Shen and co-workers, our work presents novel results and crucial advances both on the experimental and theoretical front:

We use a technique that fully quantifies the amount of non-reciprocal transmission, *i.e.*, it does not use arbitrary units for the measured transmission. This allows us to present the *first demonstration of optomechanical isolation* and quantitatively compare nonreciprocal transmission and gain to theoretical predictions. Moreover, we employ a continuous bias field, resulting in steady-state isolation (or amplification) independent of time, unlike the experiments of Shen et al. which violate reciprocity only during a short pulse.

Theoretically, we present a new general formalism of optomechanical nonreciprocity which goes beyond the interpretation in terms of the handedness of fields in ring resonators, as employed by Shen et al. (and previously by Hafezi and Rabl, ref. [27]). Importantly, our explanation directly reveals that the effect in ring resonators is *identical to a nonreciprocal phase during mode transfer*, a mechanism that has been compared to a synthetic magnetic field in other contexts [Nat. Photon. 6, 782–787 (2012); Nat. Photon. 8, 701 (2014); Nat. Commun. 5, 3225 (2014)]. The generality of our theory explains why the effects survive in absence of degeneracy (Fig 4), and it provides guidelines to extend optomechanical nonreciprocity to a variety of other optomechanical systems, such as photonic crystal resonators and superconducting LC circuits (Fig 5).

We have strengthened the explicit description of these experimental and theoretical advances with respect to other recent work on page 3 and 8 in the revised manuscript.

1. The authors suggested that their system can function as a nonreciprocal phase shifter. It will be helpful if they can present a figure showing the experimental results of nonreciprocal phase response.

We thank the referee for this suggestion. A figure showing asymmetric phase-response has been added to figure 4.

2. In the bottom of page 5, the authors have some discussion about cooperativity. The difference between forward/backward transmissivities with respect to cooperativity was presented in Figure 3a.

However, I couldn't find the particular definition of cooperativity in the manuscript, which makes it difficult to understand the relationship between the cooperativity and the forward/backward transmission.

Just above equation 10 in the main text, the cooperativity C_j is introduced and defined, where the subscript j refers to the cooperativity of the individual modes 1 and 2. This yields a total cooperativity $C = 2C_1 = 2C_2$ for degenerate modes, as was mentioned in page 5.

To clarify the use of the word cooperativity, we added the words "individual" and "total" to the respective definitions, explicitly defined the total cooperativity as " $C \equiv C_1 + C_2$ ", and changed the label of the x-axis of Fig 3a from 'cooperativity' to "Total Cooperativity".

3. The results presented in Figure 3 and Figure 4 are confusing. Figure 4b shows that optomechanically induced absorption appears when the control frequency is fixed at the "blue mechanical sideband". However, a different statement was presented in Figure 3c, which demonstrates that "when the control beam is tuned to the blue-sideband of the cavity, it can parametrically amplify the probe beam that co-propagates with it through the fibre."

We thank the referee for pointing out this potential source of confusion. The probe transmission with a blue-detuned control beam can in fact exhibit induced absorption, induced transmission, and gain, depending on the ratio $\eta = \kappa_e/\kappa$ and on total cooperativity C . In Fig 3c, η approaches 0.5, meaning near unity absorption and explaining the absence of an induced absorption peak. In contrast, in Fig 4b the system is operated in the 'undercoupled' regime ($\eta < 0.5$), which allows induced absorption. A further increase in control power would eventually also lead to a net gain, just as in figure 3c.

To avoid confusion, the text discussing Fig 4a-b on page 7 has been rewritten, explaining more carefully the different possible regimes.

4. Figure 4b shows that oppositely propagating probe experiences increased transmission. However, the results in previous figures indicate that the coupling between counter-propagating probe-control beams with the mechanical mode is negligible. Then what does the change in the probe come from if it propagates in opposite direction from the control beam?

Importantly, for nondegenerate modes (the subject of Fig 4) the *intracavity* probe field is no longer a wave propagating in the same direction as the *incident* probe field: a probe tuned to the middle of both modes will excite a superposition of the even and odd modes. If they are degenerate, it excites the modes with a $\pi/2$ phase difference, corresponding to a co-propagating superposition field in the cavity. But if the modes are split, the finite (and opposite) detuning of the probe with respect to either mode adds opposite phases to each of them. For large splitting, this approaches $\pm\pi/2$, such that the even and odd modes are excited with a $-\pi/2$ phase difference instead: counterpropagating to the incident field. At first sight this may seem to violate momentum conservation, but of course it doesn't: one should consider the fact that also reflected and transmitted outgoing waves are present in the waveguide.

In conclusion, for split modes an incident probe excites a finite *counterpropagating* intracavity field that can excite the mechanical mode by beating with a counterpropagating pump field (which has the same detuning with respect to both modes). Interestingly, this can be understood straightforwardly in the picture of mode conversion as well: For nondegenerate modes, the finite and opposite probe detuning to both modes adds an extra and opposite phase to the signals converted from a_1 and a_2 , respectively, which changes the character of their interference with the directly transmitted field. Crucially, the fact that the *control* fields of even

and odd modes are always $\sim\pi/2$ out of phase in this setup guarantees maximum isolation for *any* splitting, in line with equation 10.

To highlight this argument, the text discussing figure 4a-b has been rewritten, now including the above argumentation. In the supplementary information we note the fact that the superposition handedness depends on mode splitting when deriving the D matrix.

Reviewer #2 (Remarks to the Author):

(1) The basic scheme of optomechanical isolation presented here strikes me as very similar to the Bahl work (reference 10 in the manuscript). Bahl's paper uses electrostriction to set up OMIT for a probe wave coupling to a silica microsphere resonator. As in the present manuscript the OMIT allows passage of the probe in one direction (determined by the control) while the reverse direction is attenuated because selection rules prevent OMIT from occurring. In the Bahl paper, Brillouin phonons mediate the process in the silica microsphere via electro-striction while in the present case a mechanical breathing mode mediates the process in the micro toroid via radiation pressure. The parallel between the electro-striction-based optomechanical coupling and the radiation pressure optomechanical coupling is well known for cooling (see Bahl, ... Carmon, Nature Physics (2012)). In short, even though the work presented is sound, I am not convinced that the present demonstration is distinct enough from the Bahl nonreciprocal work to merit publication in a Nature journal. The authors should provide some comment here to answer my concern. Perhaps there is also a way to address it further in the manuscript.

The paper by Bahl and coworkers forms an important context for our work. We believe our work however presents crucial advances in both experimental demonstrations and understanding of the responsible mechanisms. Like the work of Shen et al. (see our answer to ref. 1 above), the experiments of Bahl are not quantitative, and do not report isolation, nonreciprocal gain, and performance without degeneracy, all of which we demonstrate in our paper. Importantly, our generalized theory in terms of nonreciprocal mode transfer provides a crucial advance in understanding not only our own work, but also that of Bahl et al., describing it in a new light. Whereas their interpretation focused on the fact that the employed mechanical modes exhibit angular momentum, we show that such angular momentum is not a prerequisite: *any* system with two properly driven modes jointly coupled to a mechanical mode can exhibit large nonreciprocal response and isolation. Our general multimode framework extends the concept of nonreciprocal transmission also to other optomechanical systems, such as photonic crystal resonators and superconducting LC circuits, as we discuss in the manuscript.

In the revised manuscript, we have explicitly compared our general framework to the interpretation of Bahl et al, and highlighted the experimental and theoretical advances in the context of their work.

(2) I have a concern about the practical application of the present idea for isolation. The introductory paragraph partially motivates this work on the basis of chip-based isolation and inherent limitations of present isolators. However, the present approach has two very serious weaknesses relative to commercial isolators based on magnetic materials. First, it requires a control wave that has a well-defined frequency relative to the probe wave. Second, and even more restrictive is that the isolation bandwidth is extremely narrow. Could the authors provide an example in the discussion section of the manuscript to illustrate how the present method of isolation could be applied in some way.

We thank the referee for his/her comments. We agree that our specific demonstration is narrowband and requires an external pump beam, but this criticism may be extended to the entire field of optomechanics-based nonreciprocal devices, or more generally to many papers in the recent literature exploring magnet-less realizations of isolators and circulators. There are several advantages offered by this platform, namely the possibility of integrating the device into a more complex nanophotonic system, and performing low-noise nonreciprocal transmission, ideal for quantum optics applications. In addition, the bandwidth of operation can be largely extended using different geometries. In this sense, we have added the following text to the discussion:

“(…) In fact, the high (GHz) frequency of such devices has the prospect of enhancing the bandwidth with respect to the relatively narrow range demonstrated here, towards a range commensurate with typical signal modulation rates. While the resonant nature of the demonstrated mechanism is a limit to the general application capability of our platform, we foresee several applications that may benefit from magnetic-free isolation over a finite bandwidth. These include in particular the protection of on-chip monochromatic laser sources and “(…)” the low-loss routing of signals carrying quantum information, either at optical or microwave frequencies [Metelmann/Clerk, Hafezi/Rabl].”

As an additional question, is there a geometry in which a circulator (also mentioned in the introduction) could be realized.

As first noted by C.L. Hogan (Bell Syst. Tech. J. 31, 1 (1952)), a circulator can be very generally realized by placing a gyrator in one arm of a Mach-Zehnder interferometer, where one of the four ports is reflective. The resulting three port device could thus directly be realized using a resonantly coupled optomechanical design (new figure 5a, old figure 4c) placed in a Mach-Zehnder interferometer.

We have added a sentence to the manuscript to explain the route towards constructing a circulator based on the mechanism we demonstrate.

(3) There are several critical, early works related to the present work that have unfortunately not been cited.

The referee rightfully points out the relevance of several early works, in particular 4 specific articles on (radiation pressure effects in) microtoroid resonators. Following the request of the referee, we cite all of these now in our main text to highlight the importance of these works.

Reviewer #3 (Remarks to the Author):

This manuscript presents an experimental implementation of a method to generate an effectively non-reciprocal optical system by exploiting the optomechanical interaction between a mechanical resonator and two optical fields. This contrasts with the more common mechanism used to generate optical non-reciprocity in that it does not require the use of any magnetic fields, making it inherently more suitable for chip-scale devices.

I am confident that the experimental results in the manuscript and their interpretation are broadly correct. Moreover, the topic is of great importance nowadays, as evidenced by another recent preprint [arXiv:1608.03620] that describes the experimental realisation of a similar mechanism. For these reasons I believe that the manuscript is, /in principle/, publishable in Nat. Commun. However, I

am not convinced by the mathematical methods used by the authors, as I discuss in detail below, and would like to ask them to revisit their work before resubmitting.

(1) My main technical issue with the manuscript centres around Eqs. (3) and (4). As a quantum optician, my first thought when I read the manuscript and supplementary information was that the matrices C and D can be derived by means of cavity QED and well-known input--output relations. As it turns out, these relations are inconsistent with the above-mentioned equations. Specifically, D^{T} in Eq. (3) needs to be replaced by $-D^{\dagger}$ (i.e., minus the hermitian conjugate of the matrix D). On digging deeper, it looks to me like the sign can be absorbed into a redefinition of the input fields (but not the outputs); the main problem concerns the difference between the transpose and the conjugate. Upon close inspection of Eq. (2) in the authors' Ref. [1] in the supplementary information, the answer seems to be that the authors are using equations that use the "engineers' convention" for the frequency ($da/dt = i\omega a + \dots$), whereas in the manuscript the opposite convention is used ($da/dt = -i\omega a + \dots$). A derivation of D based on cavity QED and the input--output relation also seems to disagree with Eq. (S38). I encourage the authors to provide a microscopic (by which I mean: cavity QED and input--output) derivation of their C and D matrices, which I am convinced is possible, and to ensure that their mathematics is self-consistent. The model and results seem to match quite well in any case, so I do not expect major changes to result from this, but I do expect the mathematics in the manuscript and the supplementary information to be correct.

In view of the reviewer's concern, we looked in detail into the connection between the Coupled Mode Theory (CMT) we employ – well-established in the optics and engineering literature – and the quantum optics input/output formalism that is widely used in CQED. This regards principally equations (3) and (4) of our manuscript. In the following, we show how both formalisms are mathematically related, and how it is possible to go back and forth between them. As the reviewer correctly noted, this involves a redefinition of the input (or output) fields. It is precisely the allowed freedom in the choice of ports (and the phases of fields they carry) that allows both formalisms to be consistent.

In a CQED approach, the input/output relation is conventionally written as

$$s^- = \tilde{s}^+ + Da, \quad (\text{R1})$$

where for a two mode system

$$s^- = \begin{pmatrix} s_1^- \\ s_2^- \end{pmatrix}, \quad \tilde{s}^+ = \begin{pmatrix} \tilde{s}_1^+ \\ \tilde{s}_2^+ \end{pmatrix} \quad \text{and} \quad a = \begin{pmatrix} a_1 \\ a_2 \end{pmatrix}, \quad (\text{R2})$$

denote the output, input and intracavity field operators, respectively. This is for example detailed in [Gardiner & Zoller, Quantum Noise, 3rd edition, section 5.3]. Note that in Eq. (R1), an explicit choice for the phase relation between input and output fields is made: in absence of the cavity they are related by the identity operator. Instead, in CMT a specific choice of this phase is avoided by introducing the C matrix operator such that

$$s^- = Cs^+ + Da, \quad (\text{R3})$$

which is Eq. (4) in our manuscript. Note that the relation $\tilde{s}^+ = Cs^+$ thus allows transforming outputs between the two formalisms. Applying this transformation to Eq. (3) from our manuscript,

$$\frac{d}{dt} \mathbf{a} = i\mathbf{M}\mathbf{a} + D^T \mathbf{s}^+, \quad (\text{R4})$$

and using the relation $CD^* = -D$ (obtained from time-reversal symmetry in Ref. [S1]), we obtain

$$\frac{d}{dt} \mathbf{a} = i\mathbf{M}\mathbf{a} - (CD^*)^T \mathbf{s}^+ \quad (\text{R5})$$

$$= i\mathbf{M}\mathbf{a} - D^\dagger C \mathbf{s}^+ \quad (\text{R6})$$

$$= i\mathbf{M}\mathbf{a} - D^\dagger \tilde{\mathbf{s}}^+. \quad (\text{R7})$$

This is indeed precisely equivalent to the expression mentioned by the reviewer. We thus conclude that both formalisms differ only in the sense that CQED explicitly chooses a convention for the port description, fixing the phase of the incoming waves, while CMT does not.

The CMT formalism has the benefit that it allows to associate s_j^+ and s_j^- with incoming and outgoing waves in the same physical port. Considering the CQED approach (R1) and a simple waveguide, s_1^+ and s_1^- necessarily describe waves in *different* physical ports. To study nonreciprocity, it is more insightful to have these waves in the same port, as nonreciprocity is then *always* related to a difference of the off-diagonal elements of the scattering matrix, regardless of system choice. We believe this provides a rigorous connection between the CQED and CMT approaches, motivates our preference to use the latter, and fully resolves the apparent inconsistency observed by the referee.

Lastly, we note that the fact that Fan et al. use an engineering definition in [S1] is not the reason for the apparent inconsistency (as suggested by the referee). It is straightforward to show that the frequency convention (engineering or physics) does not play a direct role in these equations, other than changing the sign of the imaginary unit in M . We use the physics convention throughout this work.

To address the reviewer's concern, we have added a paragraph to the Supplementary Information that discusses the connection between CQED and CMT, as well as a sentence in the manuscript highlighting the connection. In addition a paragraph that shows how to derive the D matrix starting from equation (R7) has been added to the Supplementary Information.

(2) In Eq. (8), b is an operator. Conservation of the canonical commutation relations requires that noise terms are added to Eq. (8). Do these noise terms have any effect? If not, they should still be included in the text for correctness. These noise terms are also missing in Eq. (S10).

The referee raises an excellent point, which we had omitted in the original manuscript simply for brevity and because thermal noise plays a negligible role in our experiments: in our situation the number of noise photons detected per second (N_{noise}) is much smaller than even the smallest probe powers used. When both optical modes are degenerate, of equal linewidth and equally driven ($|g_1| + |g_2| = g$), N_{noise} can be estimated by

$$N_{\text{noise}} = \Gamma_m N_{\text{th}} \times \frac{\Gamma_{\text{cool}}}{\Gamma_{\text{cool}} + \Gamma_m} \eta \times \frac{\Delta B}{\Gamma_{\text{cool}} + \Gamma_m}, \quad (\text{R9})$$

where $N_{\text{th}} \approx \frac{k_B T}{\hbar \Omega_m}$, $\Gamma_{\text{cool}} = \frac{4g^2}{\kappa}$, and the measurement bandwidth (in our experiment the VNA bandwidth) is given by ΔB . This equation yields $N_{\text{noise}} = \frac{CN_{\text{th}}\eta}{(1+C)^2} \Delta B \approx 4 \times 10^8$ noise photons per second for the experimental system studied here. On the other hand, at 1542 nm a probe input of 15 nW corresponds to 1×10^{11} probe photons per second, such that the detected noise is maximally 0.4% of the total number of detected photons.

To address the reviewer's comment, we have added the noise term to eq.8 and S10, and modified to manuscript to argue why it is not further considered in our experiment.

I have a few other, minor, issues that would also like to see resolved:

(3) The preprint I quoted above, arXiv:1608.03620, should be referred to.

We added a sentence that refers to this work, which appeared on arXiv after our submission, and explain the connection to our work.

(4) Following Eq. (11) it is stated that $\beta \ll \Omega_m/\kappa_{1,2}$, but no justification is given for this assertion.

We have introduced an explicit justification in the revised manuscript. In this experiment, β is found to be 6 times smaller than the ratio $\Omega_m/\kappa_{1,2}$, meaning that the drive phase differs from $\pi/2$ by $\sim 1e-3 \pi$ (0.2%).

(5) In the supplementary material, Eq. (S8) presupposes that $c_{21} = c_{12}$. This is mentioned in Ref. [1] but should be clearly stated in the manuscript.

In the revised manuscript, we explicitly state that this fact follows from the symmetry of C – a general property of a reciprocal scattering matrix.

(6) Following Eq. (S10) there is a spelling mistake: "optomchanical"

The mistake has been corrected.

(7) Following Eq. (S37) the authors do not define what they mean by "such that equations (S1) and (S2) are consistent".

In general, equations (S1) and (S2) can model light transport in the system, as long as C and D can properly incorporate the geometry of the coupled waveguide-cavity system, while the conditions for energy conservation and time-reversal symmetry of the system is ensured. Our intention in the paper was to stress this point, but we understand the confusion that this may raise. For this reason, we have now removed this confusing sentence. We thank the referee for bringing this to our attention.

As is implied above, this whole discussion would be rendered moot if the authors provided a proper microscopic derivation of matrices C and D .

In connection to the first question of the reviewer, the Supplementary Information has been updated to include a derivation of the D matrix starting from equation (R7).

REVIEWERS' COMMENTS:

Reviewer #2 (Remarks to the Author):

The authors have adequately addressed all of the points in my review. I am satisfied that the paper is now suitable for publication in Nature Communications.

Reviewer #3 (Remarks to the Author):

The authors have addressed satisfactorily all my concerns as well as the other two referees'. Considering the associated modifications made to the manuscript and supplementary information, in light of the topical nature of the subject matter, my opinion is that publication of the current version of this manuscript is justified.